# Rabies Virus Variants Detected from Cougar (*Puma concolor*) in Mexico 2000–2021

**DOI:** 10.3390/pathogens11020265

**Published:** 2022-02-18

**Authors:** Fabiola Garcés-Ayala, Álvaro Aguilar-Setién, Cenia Almazán-Marín, Claudia Cuautle-Zavala, Susana Chávez-López, David Martínez-Solís, Mauricio Gómez-Sierra, Albert Sandoval-Borja, Beatriz Escamilla-Ríos, Irma López-Martínez, Nidia Aréchiga-Ceballos

**Affiliations:** 1Instituto de Diagnóstico y Referencia Epidemiológicos, Secretaría de Salud, Mexico City 01480, Mexico; fabiola.garces@salud.gob.mx (F.G.-A.); cenia.almazan@gmail.com (C.A.-M.); claudia.cuautle@salud.gob.mx (C.C.-Z.); susana.chavez@salud.gob.mx (S.C.-L.); davidson0626@hotmail.com (D.M.-S.); mauricio.gomez@salud.gob.mx (M.G.-S.); albert.sandoval@salud.gob.mx (A.S.-B.); beatrizescamilla20@gmail.com (B.E.-R.); irma.lopez@salud.gob.mx (I.L.-M.); 2Unidad de Investigación en Inmunología, Hospital de Pediatría, Coordinación de Investigación Médica, Centro Médico Nacional Siglo XXI, Instituto Mexicano del Seguro Social, Mexico City 06720, Mexico; balantiopterix@gmail.com

**Keywords:** rabies virus, spill over, wild-living animals, antigenic variant

## Abstract

In 2019, the World Health Organization (WHO) and the Pan-American Health Organization (PAHO) recognized Mexico as a country free of human rabies transmitted by dogs. Nevertheless, the sylvatic cycle remains as a public health concern in the country. Although cougars (*Puma concolor*) are not reservoirs of any rabies virus variant (RVV), these felines could act as vectors at the top of the food chain, and their relationships with other organisms must be considered important for the regulatory effect on their prey’s populations. In this study, genetic and antigenic characterization was performed on all cougar rabies cases diagnosed at the Rabies Laboratory Network of the Ministry of Health (RLNMH) in Mexico from 2000 to 2021. Samples from other species, a skunk, a horse (*Equus caballus*) (attacked by a cougar), and a gray fox (*Urocyon cineroargenteus*), were included as reference. Rabies cases in cougars were restricted to two Northern states of Mexico (Sonora and Chihuahua). Five out of six samples of cougars were RVV7 (Arizona gray fox RVV) and one from Sonora was RVV1. Interestingly, there is no evidence of RVV1 in dogs in the Northern states since the 1990s but skunk species now harbor this RVV1 in this region of the country.

## 1. Introduction

Rabies is caused by neurotropic viruses of the genus *Lyssavirus* in the family Rhabdoviridae. It is transmissible to all mammals, and it is almost uniformly fatal [1].

In Mexico, as an achievement of the programs of surveillance and control, human rabies transmitted by dogs has been absent in the country since 2005, when the last two cases occurred. On 11 November 2019, Mexico was recognized by the World Health Organization (WHO) and the Pan-American Health Organization (PAHO) for being the first country to have fulfilled the requirements to be validated as a country free of human rabies transmitted by dogs. This was due to the sustained canine vaccination campaigns and to the epidemiological surveillance system [2].

However, sylvatic rabies remains a public health concern in Mexico because of the great diversity of wild reservoirs that maintain the virus in nature [3]. Sylvatic rabies is characterized by the involvement of wildlife to maintain stable cycles of transmission over time in particular geographic areas [4].

Mexico has a diversity of wild-living mammals estimated at 564 species [5]. In 2002, 9 out 11 rabies virus variants (RVVs) reported previously in the Americas were found within the country, and 6 were directly isolated from the reservoir species including bats and terrestrial mammals [4].

To date, rabies in mesocarnivores in Mexico is consider a consequence of a large-scale geographic establishment of long-term dog-maintained rabies enzootics, having a common ancestry between dog-maintained RVVs and the emergent dog-derived RVVs maintained mainly in gray foxes and skunks and other species [6].

Wild-living carnivores such as gray foxes (*Urocyon cineroargenteus*) considered the reservoir species of the dog-derived RVV7 and coyotes (*Canis lastrans*) harbored RVV7 and RVV1. The last one is a dog-derived RVV with an antigenic reaction pattern like the dog-maintained RVV but now is enzootic in skunk species. These two variants seem to be the major lineages infecting terrestrial wildlife animals in the north of the country.

Bobcats (*Lynx rufus*) have been reported as harboring RVV7 and RVV10, the natural reservoirs of which have been identified as (South Baja California) BCS skunk (*Spilogale putorius lucasana*), whereas RVV8 showed a wide distribution in skunk species, and RVV7 is circulating throughout west-central Mexico [6].

Recently, new RVVs have been reported with atypical antigenic reaction patterns and new species involved, such as white-nosed coatis (*Nasua narica*) [7,8], and dog-derived atypical RVV has been described in the sylvatic cycle of rabies [9]. These studies have revealed the presence of several rabies transmission cycles in wildlife species in urban and rural areas of the country and the potential of new emerging reservoir species.

From 2007–2020, 407 positive cases of rabies were confirmed by Rabies Laboratory Network of the Ministry of Health (RLNMH), in wild-living mammals. Bats represented 63% and skunks 28%, and the remaining 9% is comprised of foxes, coatis, wildcats, cougars, coyotes, deer, and opossums [2]. Since there is no program for the active surveillance of rabies virus targeting wildlife, most of these cases are related to human contact.

The cougar, also known as mountain lion (*Puma concolor*), is one of the 41 species that constitute the Felidae family of the mammalian order Carnivora, being the second-largest Neotropical carnivore. In Mexico, it is considered the most widely distributed wild-living feline [10].

Although the cougar is not a rabies virus reservoir species, they can act as vectors to humans and other mammals. The only case of human rabies (a 64-year-old person) transmitted by a cougar in Mexico took place in Batopilas, Chihuahua, in 2001. The incubation period reported for this case was 28 days [11,12].

In the United States of America (USA), there was a case of rabies in cougar each decade from 1960–2000. The detected RVVs were similar to those detected in raccoon (Florida state) and skunk (California state); RVVs from the other two cases were not reported [13,14]. In Arizona state, rabies in carnivores has been documented in the south where skunks and gray foxes maintain in circulation their own specific RVV, the south-central skunk and the Arizona gray fox (RVV7), respectively [15].

In 2020, the Larimer County Department of Health and Environment (Colorado State) reported a rabid cougar that probably had been bitten by a skunk and subsequently attacked two people. The characterization of this virus is pending [16].

The aim of this study was to perform the antigenic and genetic characterization of the rabies-positive cases in cougars diagnosed at the RLNMH in Mexico from 2000 to 2021.

## 2. Results

### 2.1. Rabies Virus Diagnosis

All cougar samples included in this study tested positive by Fluorescent Antigen Test (FAT). In some cases, the fluorescence was weaker than originally described in the laboratory records, but the brain tissue was well preserved for further characterizations.

### 2.2. Antigenic Characterization

Two RVVs were detected with a reduced panel of eight monoclonal antibodies. Five out of six samples of cougars were RVV7, and one was RVV1 detected in a cougar from Arizpe, Sonora. Interestingly, there is no evidence of RVV1 in dogs in the Northern states since the 1990s, but skunks species (species are not identified) now harbored this RVV in these states (Table 1).

### 2.3. Genetic Characterization

The phylogenetic analysis was concordant with the antigenic characterization (Figure 1). Mexican samples with the Arizona gray fox RVV7 were clustered in four major groups with a geographic trend: the Alamos cluster; the Caborca cluster grouped three samples, 251MxPumaSon14, 798MxEquineSon14 and 1763MxFoxSon19 the last was isolated directly from a gray fox (*U. cineroargenteus*), which is the reservoir species of RVV7; the Rayon-Ures cluster (from Sonora); and the Chihuahua cluster. The only case that has an atypical pattern of geographic location is the case of a cougar from Hermosillo (the capital city), Sonora, in 2002 (Figure 2). It is very likely that the cougar came from another locality. Unfortunately, there is no epidemiological data available from this case in order to confirm this.

The Texas gray fox RVV was not detected in any of the samples included in this study.

## 3. Discussion

Although feline species such as the cougar have not been identified as rabies reservoirs, they could act as vectors and transmit the rabies virus to humans and other species [17]. Members of the Felidae family have the tendency of being solitary and have only one dietary category but have no other features than have been described as important to become a rabies virus reservoir species, such as phylogenetic similarity to known reservoirs, large litters (~3.5 young/litter) and early sexual maturity; these may be some reasons why no rabies reservoirs have been described for any feline species [18,19].

This could explain why only 6 cases of rabies have been identified in cougars in the last 20 years by the RLNMH. Reports of rabid cougars in the USA are also scarce [14,20].

Even though cougar is the species of wild feline more widely distributed in Mexico, historical rabies cases diagnosed at the RLNMH are limited to 2 out of 32 states in the northern part of the country.

The detection of rabies virus in cougars only in Sonora and Chihuahua (Figure 2), indicates that interspecies transmission among wild mesocarnivores and cougars is relatively a frequent event in a limited area of these states. This has been described also in Southwestern USA, the border with Mexico, considered a hotspot for rabies outbreaks in gray foxes, striped skunks (*Mephitis mephitis*), bobcats and coyotes [21].

This might be due to natural barriers, such as mountain ranges and bodies of water, which can restrict animal movements and also can slow the spread of rabies [20]. Another possibility is that artificial barriers can limit the dispersion of the rabid cougars. Roads and highways act as barriers which hinder the movement of animals and have an ecological impact [22] affecting pathogen transmission [21].

Rabies spillovers in nature are frequently reported, although not all of them occur with the same rate of frequency [23]; moreover, spatial overlap between species if they are competing for similar resources has an important potential for these events to occur [24]. Ecological factors such as population densities or lack of aggregations of individual may prevent onward transmission by reducing intra-specific contact rates [25]. This could explain why Chihuahua and Sonora are more prone to have spillovers in cougars than other regions in the country, where the population density is lower.

The fact that the most common RVV identified in cougars in this work is the Arizona Gray fox RVV7 may indicates that cougars can prey on gray foxes or that rabid foxes attack cougars, and due to the sympatric distribution of these species, encounters can be relatively frequent.

Antigenic and genetic characterization of the rabies virus isolated from wild-living animals, contribute to define the pathways of intra-species and inter-species rabies transmission. This information is essential for evaluating the risk that wild species represent for humans and their domestic animals and to assess the need for the implementation of rabies control programs in wildlife using oral vaccination, which, since the 1970s, has been shown to be the most effective method for the control of rabies in Europe, Canada and in the USA [26,27]. Ecological, virological and genomic analyses could help to anticipate future rabies virus emergence events and identify prospective new reservoir species [28].

Comparative genetic studies strongly suggested that the gray fox is the most probable reservoir of RVV7 in the northern semi-desert plateau of Mexico. In these regions, bobcat (*Lynx rufus*) is sympatric with several potential rabies reservoirs of the family Canidae, such as the gray fox, the coyote and the desert fox (*Vulpes macrotis*) [10,29].

Historically in Mexico, RVV7 has been isolated originally from gray foxes in Zacatecas and Sonora [30,31]. Recently, RVV7 has been isolated from coyotes and bobcats along the Sierra Madre Occidental within the boundaries of Sonora and Sinaloa.

In this study, the sample from a gray fox (1763foxSon19) confirms the presence of the RVV7 in the reservoir species in Sonora. Other wild-living species in which RVV7 was detected in this study was the skunk from Alamos, Sonora (representative 1765MxskunkSon13).

In 2012, a cougar from Arizpe, Sonora, (5924PumaSon12) was infected with RVV1, a dog-maintained RVV. Nevertheless, in the north of the country since 2000, RVV1 has been detected in wild species and livestock [30]. This RVV is established in an independent cycle in the wild (skunks species) given the fact that no rabid dogs have been reported in the Northern Mexico.

Some authors have hypothesized that this could be related to the domestic dog-coyote RVV enzootic cycle identified by sequence analysis of the RVV1 nucleoprotein gene isolated from samples of coyotes and dogs [31,32,33,34]. Another option proposed is that this variant is phylogenetically related with the RVV1 rabies virus that has been isolated from some skunks in California, Arkansas and Wisconsin and in foxes in Texas [4,33], which is phylogenetically distant from the RVV1 of dogs and coyotes [4]. The genetic diversity of dog-related cases suggest long-term endemic situation with multiple independent cycles of transmission.

The sample taken from a skunk (unknown species) from Sonora (1765MxskunkSon13) represents a spillover that suggests that the skunk species in this state can act as reservoirs of RVV1, which is known from previous studies [27], and the detection of RVV7 in the skunk probably represents a spillover, which could imply that skunks can act as vectors of RVV7. This is the first evidence of this interspecies transmission from gray foxes to skunks. It is possible that the maintenance of this variant is due to independent cycles in which several wild-living species of mesocarnivores are involved and spillover has occurred constantly. This fact provides evidence of two cycles of transmission in wildlife species represented by the RVV7 and RVV1 isolates.

The bobcat isolate (*L. rufus*) (2483MxlynxrufusSon13) from Sonora was antigenically and genetically homologous to rabies variant circulating in the Arizona gray fox population, indicating a wider distribution of this variant than previously reported; however, they showed no close relationship with representatives of the Texas gray fox or any skunks (*M. mephitis*) RVV of the USA [4,33].

Our results support previous studies [33,35], since we found that variants from dogs, Texas foxes, and skunks from the north central USA, all typed as RVV1, were clearly distinguished by genetic characterization.

Since 2013, the Texas gray fox variant is considered extinct, and this RVV was last detected in a cow [36]. This variant was not detected in any of the cougar samples, even the oldest positive cases. Only the state of Chihuahua shares a border with Texas, yet all the cougar cases of this study were detected closer to Arizona. This can explain why the Texas gray fox RVV was not detected in these samples.

Nevertheless, the locality of the cases was assigned mainly where the cougars were contained. The incubation period in cougars (as in other mammals including humans) may be several weeks, and hence the spillover could have taken place far away from the locality where it was detected.

However, the clades show a geographic pattern defined by locality. Interestingly, one sample from Hermosillo, Sonora (2002), is clustered within the lineage of the samples from Chihuahua.

Since this is a retrospective study, the epidemiological records from some samples are very scarce; therefore, we consider that this sample was registered in Hermosillo (the capital city of Sonora) because the Rabies laboratory from the Ministry of Health is located in this city, and it was detected in this locality. The placement of this case in the phylogenetic analysis suggests that it is highly probable that the cougar was infected in Chihuahua, or this case can be addressed to some locality at the border of these two states.

Although cougars are not rabies virus reservoir, they represent a risk of transmission for humans and animals (wild-living species and domestic). Oral vaccination of rabies reservoir species (gray foxes and skunks) in Sonora and Chihuahua states may be the most viable alternative for the control of rabies in cougars.

## 4. Materials and Methods

### 4.1. Samples

All cougar samples received in Rabies Reference Laboratory at the Instituto de Diagnostico y Referencia Epidemiologicos (InDRE) from 2000–2021 were included in this study: samples from Chihuahua (*n* = 2) and Sonora (*n* = 4) (Figure 2). In order to have a complete data set samples from these states in the analysis were included: a skunk (non-identified species) sample from Sonora (2013), a bobcat (*Lynx rufus*) sample from Ures (2013), a horse sample (*Equus caballus*) attacked by a cougar in Caborca, Sonora (2014), and gray fox *(U. cineroargenteus*) from Sonora (2019) (considered the reservoir of RVV7 in Mexico) (Table 1).

### 4.2. Caborca Case Note (2014)

The displacement of the cougar was traced, as well as the animals and the people that were attacked. The samples from 2014 were part of the same outbreak that began to be registered on 11 March 2014, when the rabid cougar was reported. Over two days, the cougar advanced through various locations including a Lienzo Charro (the arena for practicing Charreria), where it attacked a human, a dog and an equine. It was later seen at a ranch, where it attacked a pregnant mare and two bovines.

The cougar arrived at a landfill in Caborca, where it attacked a woman and was finally contained. The brain was sent for rabies testing by FAT to the Public Health Laboratory of Sonora, and as soon as it was confirmed to be positive for rabies, the sample was sent to the Rabies Reference Laboratory at InDRE for antigenic and genetic characterization (251MxPumaSon14).

After this outbreak, 32 animals were under observation, 28 dogs, 2 equines (for several injuries) and 2 bovines which had no injuries. The rabies outbreak control activities included the vaccination of 260 dogs and cats and 52 bovines.

In addition, 34 humans received post-exposure treatment. Four dogs had to be euthanized as they presented serious injuries (all negative for rabies by FAT).

The mare attacked at the Ranch after giving birth presented signs of rabies and died in May. The brains of the mare and the foal were tested for rabies: the mare was positive (798MxEquineSon14) and the foal was negative.

### 4.3. Rabies Virus Diagnosis

The rabies diagnosis was performed directly in brain tissue by fluorescent antigen test (FAT) [37] employing Anti-Rabies monoclonal Globulin Fluorescein Isothiocyanate conjugate (FITC) (Fujirebio Diagnostics Inc., PA, USA). Further confirmation tests including antigenic characterization, RT-PCR and sequencing were performed.

### 4.4. Antigenic Characterization

The antigenic characterization was performed using an indirect fluorescent antibody technique with a reduced panel of eight monoclonal antibodies (MAbs) (C1, C4, C9, C10, C12, C15, C18, C19) as previously described [33,38]. This reduced panel is able to identify 11 reactivity patterns associated with different animals involved with rabies virus maintenance and transmission in Mexico and South America [39]. Antigenic characterization was applied directly on the brain smear.

### 4.5. RT-PCR and Partial N-Gene Sequencing

Fifty milligrams of brain tissue were homogenized in a lysis buffer: Tris 1 M, NaCl 5 M, MgCl2 0.5 M and NP40 (Sigma-Aldrich, MO, USA). The buffer aids in the homogenization of brain tissues and the hypotonic lysis of the cells to free cytoplasmic RNA [40]. Total RNA was extracted using QIAamp Viral RNA Mini Kit (Qiagen, Hilden, Germany) following manufacturer’s instructions.

Partial Nucleoprotein (*N*) gene sequences from the brain samples were amplified by RT-PCR with the following primers: 550FW (5′-ATGTGYGCTAAYTGGAGYAC-3′) position: 647–666 of the genome of the Challenge Virus Strain (CVS) [41] and 304 (5′-CGCTCTAGATTGACGAAGATCTTGCTCAT-3′); (position 1514–1533) [42] generated an 886-bp amplicon overlapping on 200 nucleotides, obtaining nearly the complete codifying sequence of the nucleoprotein gene.

### 4.6. Sequencing

The DNA sequencing was performed with the BigDye Terminator v3.1 Cycle Sequencing kit^®^ employing the ABI PRISM^®^ 3130xl Genetic Analyzer (Applied Biosystems, Foster City, CA, USA) according to the manufacturer’s recommendations. Sequences obtained in both senses were edited in ChromasPro version 1.5 (Technelysium Pty Ltd., South Brisbane, QLD, Australia). Edited files were converted to FASTA format in order to be used in the phylogenetic analysis.

### 4.7. Phylogenetic Reconstruction

A database of 45 nucleoprotein sequences was edited and aligned in the MEGA X software [43]. A total of 10 sequences were obtained in this study, and 35 reference sequences were retrieved from GenBank.

The adequate model of molecular evolution (T93+G) was determined using the software JModeltest 2.1 [44]. Phylogenetic reconstruction was performed by means of Bayesian inference using BEAST 1.8.4 [45] with five independent runs consisting of 10,000,000 generations each and a burn-in of 25.

## 5. Conclusions

Evidence obtained in this study indicates that the spillover from gray foxes to cougars is more likely in Northern Mexico, but our results also suggest that skunks could act as RVV7 vectors in this region. To our knowledge, this is the first report that describes the possible role of skunks as reservoirs/vectors of two different RVVs in the north of the country. The taxonomic and/or genetic identification of the species of skunks is paramount in order to understand the role that each species plays in the wild cycle of rabies virus.

The cougar, due to its behavior, is capable of spreading the virus in several species in relatively large areas, as it was observed in the Caborca case, which involved humans, dogs, cattle and horses in 12 km around. In conclusion, wild canids and felines may represent a risk for transmitting rabies to humans, as well as for their domestic and farm animals.

## Figures and Tables

**Figure 1 pathogens-11-00265-f001:**
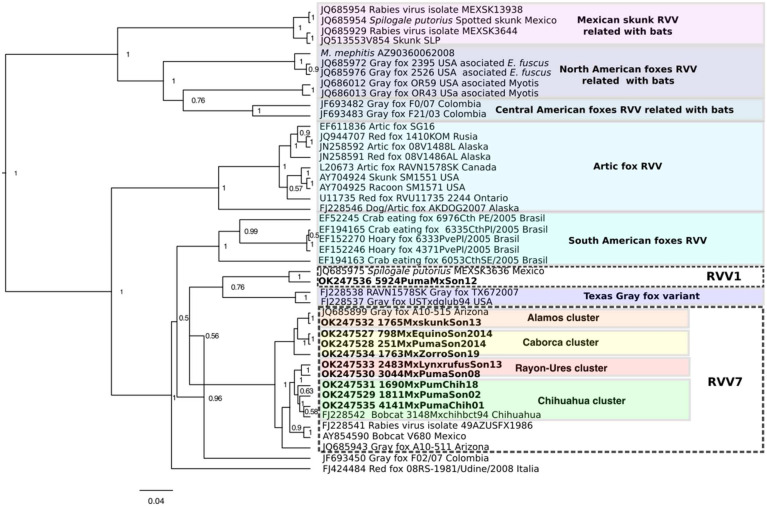
Bayesian phylogenetic tree of the rabies cases in cougar and other wild-living species of Sonora and Chihuahua including other rabies virus isolated in different reservoir species of terrestrial mammals.

**Figure 2 pathogens-11-00265-f002:**
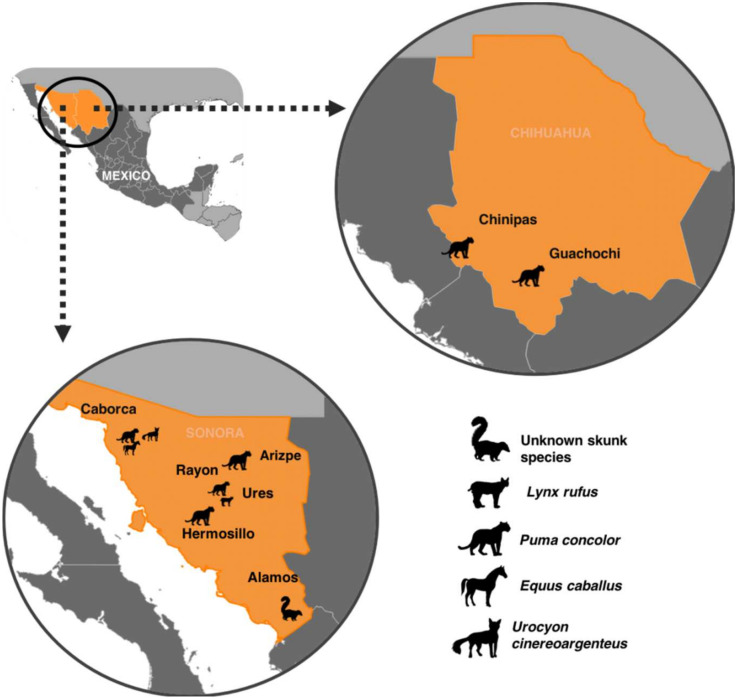
Geographical distribution of the rabies cases analyzed in this study. Localities in Sonora: Alamos, Arizpe, Caborca, Hermosillo, Rayon and Ures. Localities in Chihuahua: Chinipas and Guachochi.

**Table 1 pathogens-11-00265-t001:** Summary of cases of rabies in cougar and other species of Sonora and Chihuahua.

Case	Year	Locality	State	Species	Antigenic Rabies Virus Variant	GenBank Sequence Number
4141	2001	Guachochi	Chihuahua	*Puma concolor*	RVV7	OK247535
1811	2002	Hermosillo	Sonora	*Puma concolor*	RVV7	OK247529
3044	2008	Rayon	Sonora	*Puma concolor*	RVV7	OK247530
5924	2012	Arizpe	Sonora	*Puma concolor*	RVV1	OK247536
1765	2013	Alamos	Sonora	Skunk	RVV7	OK247532
2483	2013	Ures	Sonora	*Lynx rufus*	RVV7	OK247533
251	2014	Caborca	Sonora	*Puma concolor*	RVV7	OK247528
798	2014	Caborca	Sonora	*Equus caballus*	RVV7	OK247527
1690	2018	Chinipas	Chihuahua	*Puma concolor*	RVV7	OK247531
1763	2019	Caborca	Sonora	*Urocyon cinereoargenteus*	RVV7	OK247534

## Data Availability

All sequences generated in this study can be found at Gen Bank. Accession numbers are OK247535, OK247529, OK247530, OK247536, OK247532, OK247533, OK247528, OK24752, OK247531 and OK247534.

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
