# Peer review of "Rabies Virus Variants Detected from Cougar (Puma concolor) in Mexico 2000–2021"

_pathogens, 2022, doi:10.3390/pathogens11020265_

Round 1

Reviewer 1 Report

Spesific comments:

  1. Lines 14-17: I think the statement should be modified and shortened, ie. ‘Although pumas (Puma concolor) are not considered as reservoirs of any rabies virus variant (RVV), felines can act as vectors.’.
  2. Line 51: There appears to be a misspelling as ‘de’ probably should be ‘the’.
  3. Lines 73-74: I’m not sure what the authors want to say with this sentence, and I think the sentence should be rewritten to clarify.
  4. Line 75: It should be ‘an incubation period’ (and not ‘and incubation period’.)
  5. Lines 77-78: I think it simply could be stated that there was one case of rabies in puma in each decade rather than talking about rate.
  6. Lines 78-79: I think it should be stated that the detected RVVs were similar to those detected in raccoon and skunk.
  7. Lines 84-85: I think the last part of the sentence should be rewritten for clarification, ie. ‘…reported a rabid puma that probably had been bitten by a skunk and then attacked two people.’.
  8. Lines 90-92: It is not clear for me what the authors mean with ‘…, the antigen detected was lower than the originally described…’. Was the staining faint? Please elaborate!
  9. Lines 93-97: I don’t think this paragraph is appropriate – actually the first part of the paragraph belongs to the genetic characterization, while the last sentence could be mentioned in the introduction.
  10. Line 104: I think ‘case was reported in Hermosillo but the’ should be omitted.
  11. Line 139: I think the sentence should be reworded, ie. ‘…they could act as vectors and transmit rabies virus to humans and other species.’.
  12. Line 142: I would add ‘pumas’, ie. ‘…only six cases of rabies have been identified in pumas in the last…’.
  13. Lines 162-164: Another possibility that should be mentioned is that rabid foxes attack pumas. Anyway, it should be ‘may indicate’ (and not ‘may indicates’).
  14. Line 165: There is one space too much in ‘wild-living’.
  15. Lines 196-198: I think the sentence should be modified as the findings in this study do not support that skunks are reservoirs of RVV1. It can however be stated what is known from previous studies and that the detection of RVV7 in the skunk probably represents a spillover.
  16. Lines 208-209: This study is not about RVV in carnivores in Arizona and Texas, and I would recommend that this statement is modified as only two carnivores has been included in this study.
  17. Lines 219-222: It think it is weird to make a point of the incubation period in a human and recommend modifying the sentence just stating that the incubation period in pumas (as in other animals including humans) may be several weeks and hence the spillover can have taken place far away.
  18. Lines 233-235: This paragraph belongs to the Results!
  19. Lines 239-240: The sentence should be rewritten, ie. ‘It was later reported in a ranch where it attacked a pregnant mare and two bovines.’.
  20. Lines 236-253: These paragraphs do not belong to the Discussion – some of the information could be moved to the Introduction, and some information belong to the Materials and Methods.
  21. Line 283: It should probably be milligrams and not grams.
  22. Lines 311-314: I don’t agree that this study suggest that skunks can act as RVV1 reservoirs.

Additional comments:

  1. Some of the references are incomplete, and the authors should go thoroughly through the references!

Author Response

Dear Reviewer,

We would like to acknowledge all your comments; we consider all of them very valuable to improve the quality of this manuscript. We applied all of them unless we justified why the suggestion or comment was not considered.

Please find attached the answers.

Best regards,

Nidia Aréchiga-Ceballos

Reviewer 2 Report

In this report, the authors communicate the detection and characterization of rabies cases in cougars across two decades of passive (i.e., public health) surveillance. The case antigenic and sequence typing information is of high interest and importance because there is sparse information reported on wildlife rabies surveillance in the northern regions of Mexico, yet this region is important to understanding the limits of gray fox associated rabies viruses in the USA and presence of other wildlife rabies virus variants impacting human and animal health. The results also highlight potential for ecological interactions and rabies virus spillover from gray foxes to cougars in northwestern Mexico.

L2 – consider “rabies virus variants detected from” rather than “rabies virus variants in”; according to Wilson and Reeder mammal taxonomy, the common name is cougar. Here and throughout the paper, please consider updating for taxonomic consistency.

L15 – recommend replacing “spillovers” with “spillover infections”

L45 - recommend replacing "aerial" with "bats"

L54 - replace "in" with "infecting"; recommend replacing “wild-living animals” with “wildlife”

L46-58 – consider combining into as a single paragraph

L68 – replace “in the wild” with “targeting wildlife”

L69-75 - consider combining into as a single paragraph

L90-92 – can this be rephrased for clarity?

L109 – If the column “sequence number” pertains to Genbank, please consider a footnote?

L140-141 – please also see doi.org/10.1371/journal.pntd.0008940

L147 – please consider first reference of Figure 2 in the Results section.

L157-158 - Please consider replacing references 22 and 23 with more relevant reviews such doi.org/10.1016/j.coviro.2014.07.004 and doi.org/10.1073/pnas.2006778117

L198-199 – please clarify whether the authors mean in Mexico or otherwise which region?

L208-209 – is this an original finding of this study?

L267-269 – can the relevance of this information in the report be clarified

L270-274 - please reference the FAT method used

Author Response

(The authors gave the same response as above.)

Round 2

Reviewer 1 Report

I thank the authors for a thorough revision of the manuscript and think the manuscript may be published as it is after the 's' in 'indicates' in line 171 er deleted!